# Influence of exercise commitment on exercise adherence among undergraduate students: Chain mediation role of health beliefs and exercise behavior

JinJin Ren[1,2], PengWei Song [iD][1]*

1 Division of Physical Education, Guangxi Science and Technology Normal University, Laibin City, Guangxi Province, China, 2 Keimyung University, Daegu, Korea

* 740195450@qq.com

## Abstract

This study probed into the correlation between exercise commitment and adherence among undergraduate students and expounded on the mediating role of health beliefs and exercise behavior. A questionnaire-based survey involved 617 Chinese undergraduate students, comprising 240 males and 377 females. The results reveal a significant positive correlation between exercise commitment and adherence ($p < 0.01$). Health beliefs and exercise behavior serve as crucial mediators in this relationship. The mediation effect, which is 0.241, encompasses the individual and chain mediating effects of health beliefs and exercise behavior. The proportion of the total mediation effect attributed to each pathway is 69.7% (health beliefs), 17.4% (exercise behavior), and 12.9% (chain mediation effect). These findings offer valuable insights into enhancing exercise adherence among undergraduate students, underscoring the importance of fostering exercise commitment, cultivating health beliefs, and promoting exercise behavior. Moreover, the study provides both theoretical and practical implications for physical education reform and health promotion initiatives in universities. However, as the sample was drawn exclusively from university students in Guangxi Province, the generalizability of the findings is limited. Future studies should expand the sample scope to include a more diverse population.

## 1. Introduction

With the accelerating pace of modern life, the decline in physical function of undergraduate students has emerged as a pressing concern. Sedentary lifestyles, high academic pressure, and low levels of exercise behavior have become major factors negatively affecting students' physical health [1,2]. Although awareness of physical exercise has grown nationwide in recent years, and most university students recognize its importance, surveys indicate that undergraduate students' frequency and

**Data availability statement:** All relevant data files are available from the figshare database (accession number(s) https://doi.org/10.6084/m9.figshare.29604041).

**Funding:** This work was supported by Guangxi Science and Technology Normal University High-level Talent Research Start-up Project (GXKS2024GKY006). The funders had no role in study design, data collection and analysis, decision to publish, or preparation of the manuscript.

**Competing interests:** The authors have declared that no competing interests exist.

consistency of participation in exercise behavior remain generally low, exhibiting a common pattern of "enthusiastic participation but poor adherence [3,4]." Insufficient exercise adherence has thus become a critical issue that needs urgent resolution.

In addition, prior research has shown that regular exercise behavior significantly reduces the risk of chronic conditions such as obesity and metabolic syndrome, prompting recommendations for healthcare providers to pay closer attention to maintaining consistent exercise behavior levels among undergraduate students [5]. Santana et al. further reported that undergraduate students who did not meet the World Health Organization's (WHO) guideline of at least 150 minutes of moderate-to-vigorous exercise behavior (MVPA) per week scored higher on indicators of anxiety, depression, and reduced quality of life compared to their peers who did meet the guideline [6].

However, most existing studies have focused on the influence of single factors, lacking a comprehensive examination of the complex interplay among multiple variables. In particular, the roles of exercise commitment, health beliefs, and exercise behavior in promoting exercise adherence among undergraduate students remain underexplored. Furthermore, limited research has addressed how psychological factors influence exercise adherence through a cognitive-behavioral chain. This is especially relevant for the undergraduate population, where many students initially demonstrate high motivation for exercise but gradually reduce their engagement due to a lack of intrinsic drive, leading to decreased health levels and elevated health risks.

In response to this issue, this study delved into the correlation between exercise commitment and adherence among undergraduate students, focusing on the chain mediating role of health beliefs and exercise behavior. The findings provide scientific insights for improving exercise behaviors and offer both theoretical and practical significance in promoting students' physical and mental health and fostering healthy lifestyles.

## 1.1. Theoretical framework

This study centered on four concepts: exercise commitment, health beliefs, exercise behavior, and exercise adherence. Exercise commitment refers to an individual's psychological dedication and determination to engage in regular exercise behavior, reflecting emotional attachment to and value identification with exercise [7]. Health beliefs denote an individual's subjective cognition regarding the value of health and the outcomes of exercise, including perceived benefits of exercise behavior and awareness of the risks associated with physical inactivity [8]. Exercise behavior represents the actual participation of individuals in physical exercise [9]. Exercise adherence is defined as an individual's capacity and willingness to consistently engage in exercise behavior over time, essentially reflecting the extent of sustained involvement and execution of exercise behavior [10].

Social Cognitive Theory and the Health Belief Model constitute this study's theoretical foundation. Social Cognitive Theory emphasizes the dynamic interaction among cognition, behavior, and environment [11], offering critical insights into the formation

and maintenance of exercise behavior. The Health Belief Model highlights the role of perceived health threats and assessments of preventive actions in health behavior decision-making [12]. The effective integration of the two provides a more comprehensive explanation of the cognitive-behavioral mechanisms underlying individual actions. The establishment of exercise commitment reflects an individual's stronger pursuit of and belief in health, while health belief, in turn, effectively promotes positive social-cognitive processes that reinforce exercise behavior. Together, these mechanisms render a robust framework for exploring the relationships among exercise commitment, health beliefs, exercise behavior, and exercise adherence.

## 1.2. Literature review and research hypotheses

The concept of exercise commitment was first introduced by Scanlan et al. in 1993 to elucidate the psychological mechanisms underlying individuals' sustained participation in exercise behavior [13]. Based on Social Cognitive Theory, exercise commitment enhances individuals' self-regulatory processes, enabling them to more effectively set goals, monitor progress, and adjust behaviors [14]. Cachon-Zagalaz et al. demonstrated that individuals with higher exercise commitment exhibit stronger adherence to exercise behavior when confronted with obstacles [15]. Specifically, a stronger appreciation of the value and benefits of exercise equips individuals to better cope with challenges such as time constraints, physical fatigue, and environmental limitations, thereby fostering consistent exercise behavior engagement [16]. However, the internal mechanisms by which exercise commitment influences exercise adherence remain obscure, particularly among Chinese undergraduate students.

Hypothesis H1 is proposed:

H1: Exercise commitment can significantly and positively predict exercise adherence among undergraduate students.

According to the Health Belief Model, health beliefs encompass individuals' subjective perceptions and attitudes toward health-related behaviors. These beliefs influence exercise adherence through two primary mechanisms: first, perceived health threats—where heightened awareness of physical condition degradation or disease risk enhances motivation for behavior change [17]; second, self-efficacy—the belief in one's ability to execute behavior plays a crucial role in determining behavioral persistence [18]. Integrating the Health Belief Model with the theory of exercise commitment, it can be inferred that commitment influences individuals' cognitive appraisal of health, which in turn affects exercise adherence. Specifically, high commitment levels may lead individuals to more fully appreciate the benefits of exercise behavior and to downplay perceived barriers, thereby strengthening their health beliefs. This cognitive shift subsequently translates into more sustained exercise behavior.

Hence, the second hypothesis is posited:

H2: Health beliefs serve as a mediator between exercise commitment and adherence.

Furthermore, exercise behavior constitutes a core behavioral manifestation of exercise participation. Existing research indicates that undergraduate students with higher levels of exercise commitment tend to engage in exercise behavior with greater frequency and enhanced quality [14]. Additionally, Martínez-Sánchez et al. suggested that the positive outcomes derived from participation, such as improved physical fitness and stress relief, further reinforce exercise commitment, thereby creating a positive feedback loop between motivation and behavior [19]. More importantly, active exercise behavior is not only a consequence of exercise commitment but also a direct facilitator of exercise adherence [20].

Meanwhile, goal setting and self-monitoring promote exercise adherence. Through self-regulation, goal formulation, and feedback, individuals continually adjust and reinforce their exercise behavior, fostering stable and enduring behavioral patterns. Research has shown that those with higher exercise commitment are more adept at setting personal goals and optimizing their exercise regimens accordingly [21].

Therefore, hypothesis H3 is postulated:

H3: Exercise behavior mediates the correlation between undergraduate students' exercise commitment and adherence.

### 1.3. Overall hypothetical model

According to Social Cognitive Theory, shifts in beliefs and cognition typically precede behavioral change, while the persistence of behavioral practice contributes to the consolidation of habits [22].

In this paper, exercise commitment is assumed to initially influence individuals' health belief systems. These cognitive transformations subsequently inform behavioral intentions and promote engagement in exercise behavior, gradually shaping consistent behavioral patterns. Prior empirical research has demonstrated that the initiation of exercise behavior can generate both physiological and psychological reinforcement, forming a self-sustaining feedback loop that further consolidates habitual behaviors [23].

This belief–behavior–habit transformation chain implies that health beliefs serve as a prerequisite for exercise commitment to influence exercise behavior, while exercise behavior provides the behavioral basis for the development of exercise adherence. This sequential pathway reflects a comprehensive mechanism by which internal motivation is translated into sustained health behavior.

Consequently, hypothesis H4 is posited:

H4: Health beliefs and exercise behavior jointly mediate the correlation between exercise commitment and adherence among undergraduate students.

The model is illustrated in Fig 1.

## 2. Methods

### 2.1. Survey subject

A stratified cluster sampling method was employed. From November 5 to December 5, 2024, students were recruited from Guangxi Science and Technology Normal University, Guangxi Normal University, and Nanning Normal University. A total of 756 questionnaires were distributed, with 688 returned and 617 deemed valid, yielding a response rate of 91.0% and a valid response rate of 89.7%. The participants consisted of 240 males and 377 females. Among them, 302 were from humanities majors and 315 from science majors, with 160 freshmen, 188 sophomores, and 269 juniors. The higher

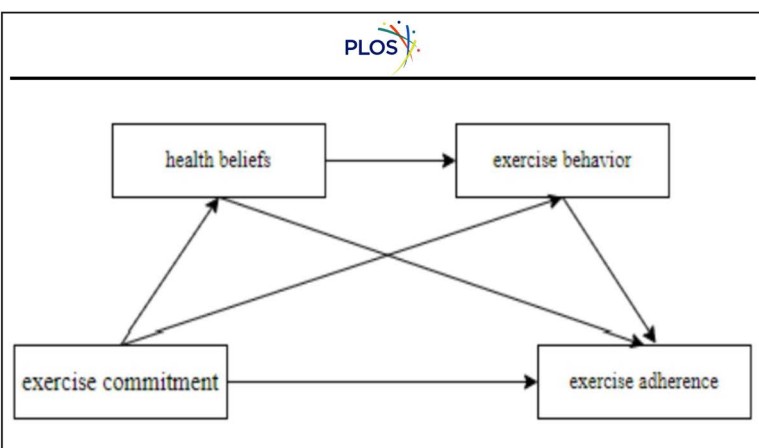

**Fig 1. Conceptual model.**

proportion of female students was attributed to the sampling universities being teacher-training institutions. Senior students were excluded from the sample as they were off-campus for internships.

This study was approved by the Ethics Committee of the School of Physical Education, Guangxi Science and Technology Normal University (Approval No: GKSTY-2024032). All participants provided informed consent.

## 2.2. Instruments

Exercise commitment, exercise adherence, health beliefs, and exercise behavior were assessed using four validated scales. Exercise behavior refers to an individual's engagement in physical activities. Exercise adherence denotes the ability and willingness to maintain such behavior over an extended period. In essence, it represents the sustained manifestation of exercise behavior. All items were rated on a 5-point Likert scale (1 = "Strongly Disagree" to 5 = "Strongly Agree"). Greater scores represent more pronounced traits or behaviors.

**2.2.1. Exercise Commitment Scale.** The Exercise Commitment Scale, adapted from Ling and A [24], consists of 14 items across three dimensions: enjoyment of exercise, participant opportunities, and personal dedication. This scale assesses the degree of psychological commitment to exercise behavior. The Cronbach's α was 0.910.

**2.2.2. Exercise Adherence Scale.** The Exercise Adherence Scale compiled by Jiang and Y [25] includes 14 items covering three dimensions: endeavor investment, behavioral habit, and emotional experience. It evaluates the degree of persistence in exercise behavior. The scale had a Cronbach's α of 0.944.

**2.2.3. Health Belief Scale.** The Health Belief Scale developed by Zhang and L [26] comprises 24 items measuring five dimensions: perceived benefits of exercise, self-efficacy in fitness evaluation, susceptibility to illness based on fitness levels, perceived severity of disease and weakness, and concern about fitness evaluation outcomes. The scale's Cronbach's α was 0.966.

**2.2.4. Exercise Behavior Scale.** The Exercise Behavior Scale, revised by Zhao and Y. [27], contains six items across two dimensions: exercise intensity and subjective perception of exercise behavior. Responses were scored from 1 (light activity) to 5 (vigorous activity). The Cronbach's α was 0.842.

## 2.3. Statistical analysis

Data were analyzed using SPSS 23.0 and the PROCESS macro (Model 6) developed by Hayes. Analyses included tests for common method bias, assessment for reliability and validity, descriptive statistics, Pearson correlations, and regression analyses. The Bootstrap method was employed to test mediation and chain mediation models among exercise commitment, exercise adherence, health beliefs, and exercise behavior. The significance level was set at α = 0.05.

# 3. Results

## 3.1. Common method bias test

To ensure the effective administration of the survey and to mitigate potential common method bias, this study employed a fully anonymous questionnaire design, encouraged participants to provide sincere responses, and facilitated data collection in a quiet environment. These precautions substantially reduced the likelihood of common method bias.

Common method bias was assessed using Harman's single-factor test. The 59 measurement items in the four scales were subjected to exploratory factor analysis. Factors with eigenvalues exceeding 1 were extracted. Unrotated principal component analysis indicated that the first factor accounted for 36.21% of the total variance, below the critical threshold of 40% [28]. This suggests that common method bias was not a major concern.

## 3.2. Descriptive statistics and correlation analysis

After controlling for demographic variables, Pearson correlation and descriptive statistical analyses were conducted for all variables, with the results presented in Table 1. Exercise commitment among undergraduate students was significantly

and positively correlated with health beliefs (r=0.591, p<0.01), exercise adherence (r=0.446, p<0.01), and exercise behavior (r=0.334, p<0.01). Additionally, health beliefs were positively associated with both exercise adherence (r=0.491, p<0.01) and exercise behavior (r=0.351, p<0.01), while a significant positive correlation was also observed between exercise adherence and exercise behavior (r=0.396, p<0.01).

### 3.3. Mediation analysis of health beliefs and exercise adherence

This study employed Hayes' PROCESS macro to test for mediation effects, as it is well-suited for analyzing complex multiple mediation models and provides accurate estimates of indirect effects. The significance of mediation was assessed using the Bootstrap resampling method with 5,000 samples and a 95% confidence level. If the confidence interval does not include zero, the mediation effect is considered significant. Regression analysis revealed that exercise commitment significantly and positively predicted health beliefs (β=0.591, p<0.01) (Table 2). This finding validates the first path in the mediation model. Both exercise commitment and health beliefs significantly and positively predicted exercise behavior (β=0.182, β=0.220, p<0.01), while exercise commitment, health beliefs, and exercise behavior all significantly and positively predicted exercise adherence (β=0.188, β=0.284, β=0.233, p<0.01).

The mediation effects were verified using the Bootstrap method, with results shown in Table 3. The 95% confidence intervals for the total indirect effects of health beliefs and exercise behavior and the indirect effects of various paths excluded zero, indicating that both individual mediation effects and the chain mediation effect of health beliefs and exercise behavior were statistically significant (Fig 2). Specifically, three significant indirect paths were identified: (1) exercise commitment→health beliefs→exercise adherence, with an indirect effect of 0.168 (39.16% of the total effect), confirming that health beliefs play an essential mediating role and H2 is solid; (2) exercise commitment→exercise behavior→exercise adherence, with an indirect effect of 0.042 (9.79% of the total effect), indicating that exercise behavior also serves as a key mediator, supporting H3; and (3) exercise commitment→health beliefs→exercise behavior→exercise adherence,

**Table 1. Descriptive statistics and correlation analysis of variables (M±SD).**

| Variable | M±SD | 1 | 2 | 3 |
|---|---|---|---|---|
| Exercise commitment | 3.41±0.84 | 1 | | |
| Health beliefs | 3.78±0.85 | 0.591** | 1 | |
| exercise behavior | 3.13±0.79 | 0.334** | 0.351** | 1 |
| Exercise adherence | 3.51±0.81 | 0.446** | 0.491** | 0.396** |

Note: *P<0.05, **P<0.01, ***P<0.001.

**Table 2. Regression analysis between variables.**

| Regression Equation | | Model Fit Indicator | | | Regression Coefficient | | Confidence Interval | |
|---|---|---|---|---|---|---|---|---|
| Dependent Variable | Independent Variable | R | R² | F | β | t | LLCI | ULCI |
| Health Beliefs | Exercise Commitment | 0.591 | 0.350 | 330.83** | 0.594 | 18.18** | 0.530 | 0.658 |
| Exercise Behavior | Exercise Commitment | 0.385 | 0.148 | 53.31** | 0.182 | 4.22** | 0.097 | 0.267 |
| | Health Beliefs | | | | 0.220 | 5.10** | 0.135 | 0.304 |
| Exercise Adherence | Exercise Commitment | 0.568 | 0.323 | 97.27** | 0.188 | 4.67** | 0.109 | 0.266 |
| | Health Beliefs | | | | 0.284 | 7.03** | 0.204 | 0.363 |
| | Exercise Behavior | | | | 0.233 | 6.39** | 0.160 | 0.306 |

Note: *P<0.05, **P<0.01, ***P<0.001.

**Table 3. Mediating effect verification of health beliefs and exercise behavior.**

| Influence Pathway | Effect Size | BootSE | BootCI | Ratio of Mediation Effect |
|---|---|---|---|---|
| Exercise Commitment→Health Beliefs→Exercise Adherence | 0.168 | 0.029 | [0.11,0.228] | 39.16% |
| Exercise Commitment→exercise behavior→Exercise Adherence | 0.042 | 0.014 | [0.017,0.072] | 9.79% |
| Exercise Commitment→Health Beliefs→exercise behavior→Exercise Adherence | 0.031 | 0.010 | [0.013,0.053] | 7.23% |
| Total Effect | 0.429 | 0.035 | [0.361,0.497] | |
| Direct Effect | 0.188 | 0.040 | [0.109,0.267] | 43.82% |
| Total Indirect Effect | 0.241 | 0.031 | [0.180,0.303] | 56.18% |

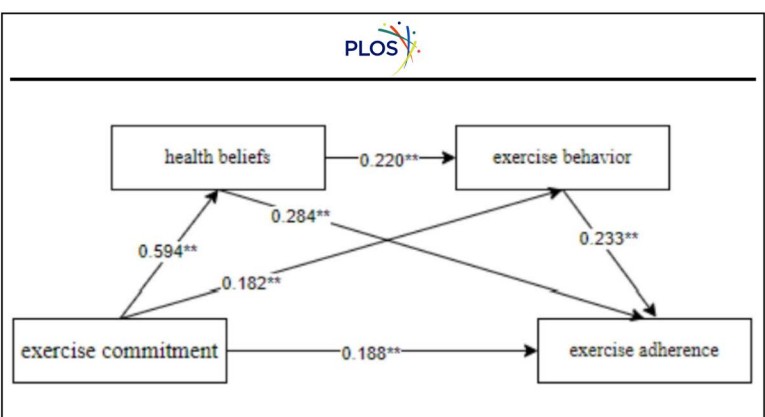

**Fig 2. Chain mediation model.**

with an indirect effect of 0.031 (7.23% of the total effect), validating H4. Overall, the total effect of exercise commitment on exercise adherence is 0.429, comprising a direct effect of 0.188 and a total indirect effect of 0.241.

## 4. Discussion

This study reveals significant positive correlations among exercise commitment, health beliefs, exercise adherence, and exercise behavior in undergraduate students. Exercise commitment directly impacts exercise adherence while affecting it via the mediating role of health beliefs and exercise behavior. This illustrates a comprehensive psychological-to-behavioral transformation mechanism from mental commitment to behavioral persistence, offering valuable theoretical insight into the internal psychological mechanisms that underpin exercise adherence.

### 4.1. Interaction between exercise commitment and adherence

Exercise commitment significantly and positively predicts exercise adherence, supporting H1. This aligns with the findings of Li et al. [29] and substantiates Cachon-Zagalaz et al.'s theoretical proposition [15] that exercise commitment is a crucial predictor of sustained exercise behavior. According to Social Cognitive Theory, exercise commitment fosters individual initiative and continuity in exercise behavior by promoting psychological needs, goal setting, and self-monitoring mechanisms [30]. Specifically, undergraduate students with higher exercise commitment are more capable of setting goals, monitoring progress, and adjusting their behaviors, enabling them to overcome barriers to exercise and maintain adherence. Notably, compared to other age groups, undergraduate students face fewer external constraints on exercise, relying more on autonomous decision-making, making exercise commitment particularly influential in this demographic.

In summary, Social Cognitive Theory underscores the pivotal role of exercise commitment as both a motivating force and a safeguard for long-term exercise participation among undergraduate students. Therefore, universities should place greater emphasis on fostering students' exercise commitment. Particularly, incentive mechanisms should be established to stimulate their motivation to participate in exercise and enhance their enjoyment of physical exercise. For example, implementing diversified sports programs, using digital self-monitoring tools to identify and reward students who maintain regular exercise, establishing a campus competition system with weekly sporting events, incorporating exercise duration into course assessments, and continuously providing positive reinforcement along with moderately challenging physical tasks can all enhance students' enjoyment of exercise, thereby promoting the sustainability and effectiveness of college students' exercise behavior.

### 4.2. Mediating effect of health beliefs

Health beliefs partially mediate the relationship between exercise commitment and adherence, asserting H2. This finding not only aligns with the core assumptions of the Health Belief Model [17,18] but also highlights the critical role of cognitive factors in shaping behavior. Consistent with Takemura et al.'s findings, health cognition serves as a pivotal intermediary linking motivation with sustained engagement in exercise behavior [31]. Additionally, Ley et al. found that self-regulation remained significantly associated with exercise behavior levels and adherence even after controlling for intrinsic motivation and positive emotions [32], reinforcing the importance of cognition and self-regulation mechanisms in exercise adherence. Specifically, individuals with strong exercise commitment tend to perceive greater control in exercise settings, which enhances their confidence in physical capabilities. This heightened sense of control fosters a more positive appraisal of exercise value, strengthening health beliefs and ultimately facilitating consistent exercise adherence. This progression illustrates a psychological transition from commitment to persistence.

Furthermore, while the traditional Health Belief Model mainly focuses on disease prevention behaviors [33,34], this study extends its relevance to exercise behavior, demonstrating the model's cross-contextual applicability. This shift aligns with the evolving health paradigm from risk prevention to proactive health promotion. Consequently, it is recommended that universities and relevant institutions prioritize the cultivation of students' health beliefs, strengthen health education and psychological support, and help students fully recognize the benefits of self-directed exercise. Additionally, the establishment of effective physical health evaluation systems should be promoted to continuously enhance students' proactive engagement, thereby improving their exercise adherence and overall health levels.

### 4.3. Mediating impact of exercise behavior

Exercise behavior partially serves as a mediator between exercise commitment and adherence, validating H3. Students with stronger exercise commitment demonstrate greater initiative and persistence in exercise behavior, and such positive behaviors further contribute to the development of adherence. This process reflects a progressive model of "commitment–behavior–adherence," depicting the trajectory from psychological motivation to habitual behavior. The result agrees with Teixeira et al.'s conclusion that autonomous motivation plays a critical role in maintaining exercise behavior [35]. Similarly, Zhang et al. showed that when basic psychological needs such as autonomy, competence, and relatedness are fulfilled, individuals generate higher-quality motivation that facilitates behavioral persistence and adherence [36]. These theoretical perspectives resonate with the findings in this study, collectively highlighting the importance of translating motivation into actual behavior in the development of exercise adherence.

Moreover, this research enriches the theoretical understanding of exercise commitment by emphasizing its role as a firm determination and sense of responsibility toward exercise behavior, which can be effectively translated into behavioral performance and ultimately enhance exercise adherence. From a practical perspective, universities should encourage students to enter into relevant sports participation agreements and employ apps to monitor their exercise volume and intensity, ensuring the occurrence of physical activity. Furthermore, by enriching the variety of sports activities and providing

ongoing incentives, universities can assist students in translating intrinsic motivation into concrete actions. Continuous participation is a pivotal element in fostering exercise adherence. It plays a critical role in promoting the physical and mental health of college students.

### 4.4. Chain mediation effect of health beliefs and exercise behavior

Health beliefs and exercise behavior play a chain mediating role in the relationship between exercise commitment and adherence, thereby confirming H4. This aligns with Rhodes et al.'s systematic review [37] while further elucidating the specific pathway of the cognition–behavior facilitation mechanism in exercise adherence. Exercise commitment functions as an internal driving force that enhances health beliefs, which in turn promote exercise behavior, ultimately strengthening adherence. This is in line with Social Cognitive Theory, which posits that both cognitive and behavioral factors facilitate the development of exercise adherence [11]. Meanwhile, it echoes the perspective that shifts in beliefs and cognition typically precede behavioral change, while sustained practice reinforces habitual behaviors [22]. Therefore, strategies to promote exercise adherence should simultaneously focus on building health beliefs and maintaining exercise behavior engagement. Particularly, given that health beliefs serve as the strongest mediating variable, intervention measures aimed at improving college students' exercise adherence should place greater emphasis on enhancing students' self-efficacy in physical health evaluations, awareness of the benefits of regular exercise, and perception of the severity of illness and physical weakness, thus fostering their habitual engagement in exercise.

However, it is worth reflecting that while this study explains the cognitive-to-behavioral transformation at the individual level, it may overlook the influence of broader social-environmental factors. From a social cognitive perspective, exercise adherence is also shaped by interpersonal relationships, social support, resources, and policy contexts [38,39]. Future research should integrate individual psychology with environmental factors to build a more comprehensive theoretical model of exercise adherence, thereby offering stronger foundations for practical interventions and policy development.

## 5. Conclusions and limitations

This study discloses significant positive correlations among exercise commitment, health beliefs, exercise behavior, and exercise adherence in undergraduate students. Exercise commitment directly influences exercise adherence while impacting it via the individual mediating roles of health beliefs and exercise behavior, as well as a chain mediating pathway involving both. This reveals a comprehensive psychological-to-behavioral transformation mechanism from commitment to sustained behavior, offering theoretical insights and practical guidance for physical education reform and health promotion initiatives targeting undergraduate students. However, since the study merely incorporated university students from Guangxi Province, the generalizability of the findings is constrained.

Although this study examines the relationship between college students' exercise commitment and exercise adherence from the perspectives of social cognitive theory and the health belief model, as well as the individual and serial mediating effects of health beliefs and exercise behaviors, several limitations remain:

First, this study employs a cross-sectional design to explore the relationships between variables. While the study confirms the associations between the results, it does not establish causal paths and thus cannot make causal inferences. This cross-sectional design fails to elucidate the causal relationships among the variables. The temporal sequence of relationships among exercise commitment, health beliefs, sports behaviors, and exercise adherence remains unclear. This design hinders a comprehensive understanding of the underlying mechanisms. Longitudinal data would more effectively track the relationships between exercise commitment, health beliefs, sports behaviors, and exercise adherence over extended periods. Future research should conduct follow-up surveys or experimental studies to investigate how these variables influence college students' physical behavior and exercise adherence over the long term.

Second, the sample in this study is limited to the Guangxi Autonomous Region. Regional differences in lifestyle, cognition, and geographical environment may lead to variations in exercise commitment, health beliefs, sports behaviors, and

exercise adherence. This suggests that the findings may not be entirely applicable to college students in other regions. The data obtained may degrade the generalizability of the results. Subsequent studies should broaden the sample scope and consider potential moderating variables (such as social support and environmental factors) that may influence the mediating pathways.

Third, although the serial mediation effect is significant, there may be a bidirectional relationship between health beliefs and sports behaviors. To gain a more comprehensive understanding of the mechanisms by which health beliefs and sports behaviors influence each other, future studies should delve into this dynamic process.

## Author contributions

**Conceptualization:** PengWei Song.

**Data curation:** JinJin Ren.

**Formal analysis:** PengWei Song.

**Investigation:** JinJin Ren.

**Methodology:** JinJin Ren, PengWei Song.

**Project administration:** JinJin Ren.

**Resources:** JinJin Ren.

**Software:** PengWei Song.

**Validation:** PengWei Song.

**Writing – original draft:** JinJin Ren.

**Writing – review & editing:** PengWei Song.

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
