## [Decision Letter · Decision Letter 0]

18 Jul 2025

Dear Dr. song,

Thank you for submitting your manuscript to PLOS ONE. After careful consideration, we feel that it has merit but does not fully meet PLOS ONE’s publication criteria as it currently stands. Therefore, we invite you to submit a revised version of the manuscript that addresses the points raised during the review process.

**ACADEMIC EDITOR:** Thank you for your submission. In order to proceed further with the editorial process and prepare your manuscript for official publication, we kindly request that you address all reviewer comments thoroughly and revise the manuscript accordingly. Please ensure that your responses are clear, well-structured, and that all requested clarifications, additions, or corrections are incorporated.

We look forward to receiving your revised manuscript.

Kind regards,

Nadia Rehman, Ph.D.

Academic Editor

PLOS ONE

Journal Requirements:

4. We are unable to open your Supporting Information file [data.sav]. Please kindly revise as necessary and re-upload.

5. Please include captions for your Supporting Information files at the end of your manuscript, and update any in-text citations to match accordingly. Please see our Supporting Information guidelines for more information: http://journals.plos.org/plosone/s/supporting-information .

Reviewers' comments:

Reviewer's Responses to Questions

**Comments to the Author**

1. Is the manuscript technically sound, and do the data support the conclusions?

Reviewer #1: Yes

Reviewer #2: Yes

Reviewer #3: Yes

Reviewer #4: Partly

2. Has the statistical analysis been performed appropriately and rigorously?

Reviewer #1: No

Reviewer #2: Yes

Reviewer #3: Yes

Reviewer #4: Yes

3. Have the authors made all data underlying the findings in their manuscript fully available?

Reviewer #1: No

Reviewer #2: Yes

Reviewer #3: No

Reviewer #4: Yes

4. Is the manuscript presented in an intelligible fashion and written in standard English?

Reviewer #1: Yes

Reviewer #2: Yes

Reviewer #3: Yes

Reviewer #4: Yes

Reviewer #1: The study investigates an important and timely topic: the mechanisms through which exercise commitment influences exercise adherence among undergraduate students, with a focus on the chain mediating roles of health beliefs and exercise behavior. The research question is highly relevant to the fields of health psychology and exercise science. Overall, the manuscript is well-structured, the theoretical framework is sound, and the use of a chain mediation model (PROCESS Model 6) is appropriate for the stated hypotheses.

While the study has clear merits and potential for publication, I have identified several issues that must be addressed before it can be considered further. My detailed comments are provided below.

General Comments: The manuscript successfully tests a conceptual model that explains how psychological commitment is translated into sustained behavior. The finding that exercise commitment influences adherence through direct, indirect (via health beliefs and exercise behavior separately), and chain mediation pathways offers a nuanced contribution to the literature. The study's strengths include its relevant topic, a large sample size, a clear theoretical foundation in Social Cognitive Theory and the Health Belief Model, and appropriate statistical methods. However, the manuscript suffers from critical inconsistencies in administrative declarations, errors in referencing, and a need for deeper interpretation of the findings and limitations.

Major Issues

1.Critical Contradiction in Funding Disclosure: There is a major discrepancy regarding the funding source. The online submission information states: “The author(s) received no specific funding for this work.” However, the manuscript text under the "Funding" section explicitly lists: “Guangxi Science and Technology Normal University High-level Talent Research Start-up Project (GXKS2024GKY006).” This is a serious contradiction that undermines the manuscript's integrity. The authors must clarify which statement is correct and ensure all information is consistent across all submitted materials.

2.Critical Contradiction in Data Availability Statement: A similar contradiction exists for data availability. The submission system states: “All relevant data are within the manuscript and its Supporting Information files.” However, the manuscript text under "Data available" states: “Data is available from the corresponding author on request.” The authors must resolve this discrepancy and provide the data without restriction (e.g., as a supporting information file), ensuring the statement is accurate and uniform.

3.Clarification Required for Reference [5]: This citation requires clarification to ensure academic rigor. Reference [5] points to a University of Iowa news article with a listed publication date of January 2025. It is understood that literature can be incorporated into a manuscript after the initial data collection is complete (which occurred in late 2024). However, citing a source with a future publication date is unconventional. For transparency and verifiability, the authors should clarify the status of this source. Was it an advance online publication or an in-press article at the time of writing? Furthermore, while news articles can provide context, citing the original peer-reviewed study being reported on would significantly strengthen the manuscript's scientific foundation. We recommend the authors replace this reference with the primary research article if possible, or at minimum, correct the citation to reflect its precise publication status and provide a DOI if available.

4.Insufficient Discussion of Cross-Sectional Design Limitations: The authors acknowledge the cross-sectional design as a limitation but do not adequately discuss its implications. For a model testing a causal chain (Commitment → Beliefs → Behavior → Adherence), the inability to establish temporal precedence is a fundamental weakness. The discussion should be expanded to explicitly state that the findings represent associations rather than proven causal pathways. The authors should be more cautious with their language throughout the manuscript to reflect the correlational nature of their data.

5.Results Section: On lines 235-236, the text states: "exercise commitment significantly and positively predicted health beliefs (β=0.591, p<0.01), supporting H1." This is incorrect. This result supports the first path in the mediation model, not H1, which posits that exercise commitment predicts exercise adherence. Please correct this statement to accurately reflect what the result supports.

6.Discussion Section: The practical implications are somewhat generic (e.g., "optimize exercise environments"). The authors should provide more specific, actionable recommendations based on their findings. For instance, given that health beliefs were the strongest mediator (accounting for 69.7% of the mediation effect), what specific components of health beliefs (e.g., perceived benefits, self-efficacy) should university interventions target?

Reviewer #2: The study has considerable scientific merit, featuring a robust design and findings that make a significant contribution to the understanding of exercise adherence among university students. The clarity of the writing is also a notable strength.

Its strengths include:

- Rigorous and Appropriate Statistical Analysis: The statistical analysis was conducted correctly and with rigor.

- Addressing Common Method Bias: The use of Harman’s single-factor test and the finding that the first factor accounted for only 36.21% of the total variance (below the 40% threshold) is a strength, indicating that common method bias was not a significant concern.

- Controlled and Detailed Correlations.

- Hypotheses Supported by the Data: All hypotheses (H1, H2, H3, and H4) were consistently supported by the findings, with significant indirect effects and confidence intervals that excluded zero. This demonstrates the validity of the proposed relationships.

- Instrument Handling and Reliability: Validated scales were used to measure key variables, and the high reliability of these scales indicates strong data quality.

- Clarity and Comprehensibility of Language: The manuscript is presented in a clear and intelligible manner, written in standard English, with correct and unambiguous language that facilitates reading and understanding.

- Valuable Practical and Theoretical Implications: The findings provide valuable insights for improving exercise adherence among university students, emphasizing the importance of fostering exercise commitment, cultivating health beliefs, and promoting exercise behavior. The study also offers theoretical and practical implications for reforming physical education and promoting health initiatives in universities.

- Ethical Approval and Informed Consent.

Points to Note or Improve:

- Inconsistency in Funding Statement: There is a contradiction in the funding section. Initially, it states that “The authors received no specific funding for this work.” However, it later mentions specific funding: “Start-up Research Project for High-Level Talent at Guangxi University of Science and Technology (GXKS2024GKY006).” The authors must clarify and reconcile this information to ensure transparency and accuracy in the funding disclosure.

Reviewer #3: Regarding the availability of the data = Partially. The manuscript mentions that the data are available upon request from the corresponding author, which does not fully align with PLOS’s data availability policy that requires open access to underlying data without restriction (with limited exceptions). However, a supporting file (data.sav) is included as a downloadable item via Editorial Manager, which may satisfy the policy depending on its content. This should be clarified with a statement confirming the availability of raw, anonymized data behind all figures and statistical results, such as means, SDs, and individual scores.

Recommendation: The Data Availability Statement must be updated to confirm that the .sav file is public, anonymized, and sufficient for replication.

Reviewer #4: First of all, I would like to congratulate the authors for addressing such a relevant and timely topic as exercise adherence among university students. The study is well-focused, clearly states its objectives, and is appropriately grounded in a robust theoretical framework, effectively drawing upon Social Cognitive Theory and the Health Belief Model. The authors have successfully integrated theory and empirical findings, clearly formulating hypotheses supported by current and relevant literature.

I particularly commend the methodological choices made in the study, including the use of validated scales with very high internal reliability, which considerably strengthens the credibility and robustness of the findings. Additionally, the statistical analysis is rigorous and well-detailed, correctly employing Hayes’ PROCESS macro with bootstrapping for the chain mediation analysis. The interpretation of results is clear, balanced, and consistent with the reported findings.

However, from a constructive viewpoint, I would suggest addressing a few points to further enhance the quality and impact of the manuscript:

Study design: The main limitation of this study is its cross-sectional design, significantly constraining the ability to draw clear causal conclusions. Although the authors acknowledge this limitation, I would recommend emphasizing it more explicitly in the discussion. Future longitudinal or experimental studies would be very useful to better validate the causal relationships proposed.

Potential biases associated with self-report measures: While the statistical test indicated that common method bias was not dominant, it would be helpful to explicitly state the preventive measures implemented to minimize such biases (e.g., complete anonymity, clear instructions encouraging sincere responses).

Data availability: Although the authors state that data are available without restriction, it would be beneficial to specify explicitly the repository or exact platform where the dataset can be accessed. This would strengthen transparency and allow other researchers to easily replicate your analyses.

In summary, I believe this manuscript has great potential to contribute significantly to the field of promoting healthy habits among university students. By addressing these minor recommendations, the manuscript will fully meet editorial standards and provide notable value to the scientific community interested in these topics.

**Do you want your identity to be public for this peer review?** For information about this choice, including consent withdrawal, please see our Privacy Policy

Reviewer #1: No

Reviewer #2: No

Reviewer #3: No

Reviewer #4: No

---

## [Author Response · Author response to Decision Letter 1]

30 Aug 2025

Comments from the Academic Editor:

1. The article format was revised in accordance with the formatting requirements of PLOS ONE.

2. The funding disclosure was modified and stated in the Cover Letter.

3. The raw data of the current manuscript was deposited in a public resource repository, with open and unrestricted access.

4. The raw data has already been uploaded to the public resource repository.

5. The titles of the supporting information files were revised.

6. The referenced works were updated.

Comments from the Reviewers:

1. Reviewer 1’s first comment pointed out that there were significant discrepancies in the funding disclosure. The financial disclosure has now been stated in the Cover Letter. The correct financial disclosure is as follows: Guangxi Science and Technology Normal University High-Level Talent Research Start-Up Project (GXKS2024GKY006).

2. Reviewer 1’s second comment indicated inconsistency regarding data availability. The raw data of the manuscript was already deposited in a public resource repository, with open and unrestricted access. The uploaded data fully supports all research results in this study. Data access link: https://doi.org/10.6084/m9.figshare.29604041。

3. Reviewer 1’s third comment indicated that Reference 5 was a news article and an online early release, which does not meet publication requirements. We replaced it with a 2023 research article.

4. Reviewer 1’s fourth comment indicated that the authors acknowledged the limitations of the cross-sectional design but did not sufficiently discuss its impacts. The discussion was revised to more clearly highlight the shortcomings of the cross-sectional study and propose improvement strategies.

5. Reviewer 1’s fifth comment indicated that lines 235–236 stated, "Exercise commitment significantly and positively predicts health beliefs (β = 0.591, p < 0.01), which does not support Hypothesis 1." This has now been revised to: "Exercise commitment significantly and positively predicts health beliefs (β = 0.591, p < 0.01), which supports the first path in the mediation model."

6. Reviewer 1’s sixth comment indicated that the practical implications in the discussion section were unclear (e.g, “optimizing the exercise environment”). Based on the research results, more specific and feasible suggestions have now been provided.

7. Reviewer 2 indicated inconsistencies in the funding statement. The mistake in the funding section have now been clarified and corrected.

8. Reviewer 3 indicated inconsistencies regarding data availability. The accessible data has now been uploaded to the repository.

9. Reviewer 4 suggested making it clear what preventive measures were implemented to reduce such biases. The preventive measures taken to minimize common method bias have now been explicitly explained.

10. Reviewer 4 indicated that the limitations of the cross-sectional study and proposed solutions should be clearly stated. We further described the limitations of the cross-sectional study and proposed future longitudinal and experimental research in the future study.

11. Reviewer 4 requested that the dataset should be uploaded to a public repository. The raw data was already uploaded to the public resource repository.

---

## [Decision Letter · Decision Letter 1]

25 Sep 2025

Dear Dr. song,

Thank you for submitting your manuscript to PLOS ONE. After careful consideration, we feel that it has merit but does not fully meet PLOS ONE’s publication criteria as it currently stands. Therefore, we invite you to submit a revised version of the manuscript that addresses the points raised during the review process.

The manuscript demonstrates strong potential and addresses an important research problem. However, to ensure clarity and avoid overstatement, I recommend **Minor Revisions** prior to acceptance. Specifically, the authors should:

Adjust causal language to reflect the cross-sectional design.Clarify the scope of generalizability.Differentiate more clearly between “exercise behavior” and “exercise adherence.”Strengthen the theoretical integration and expand practical implications.Correct minor inconsistencies in ethics approval and references.

We look forward to receiving your revised manuscript.

Kind regards,

Nadia Rehman, Ph.D.

Academic Editor

PLOS ONE

Journal Requirements:

Additional Editor Comments:

*****The sample is restricted to universities in Guangxi Province, which may limit external validity. This should be explicitly stated in both the Abstract and Conclusion to avoid overgeneralization to all undergraduates.

****(Exercise Behavior vs. Exercise Adherence):While measured with different instruments, the conceptual distinction between these constructs could be further clarified in the Methods.

****The manuscript applies both Social Cognitive Theory and the Health Belief Model but largely in parallel. Greater emphasis on how their integration offers novel explanatory power would strengthen the contribution.

****Recommendations such as “incentive mechanisms” remain vague. Expanding this section with concrete examples (e.g., digital self-monitoring tools, peer-led initiatives, or curriculum-based interventions) would enhance applied value.

****Some foundational citations are dated (e.g., Glanz, 1997). Consider balancing with more recent literature (2022–2024) to reflect current advances. Read and cite https://doi.org/10.1186/s40359-025-02767-0 ; https://doi.org/10.1108/QEA-06-2024-0054

Reviewers' comments:

Reviewer's Responses to Questions

**Comments to the Author**

Reviewer #1: All comments have been addressed

Reviewer #2: All comments have been addressed

Reviewer #4: All comments have been addressed

2. Is the manuscript technically sound, and do the data support the conclusions?

Reviewer #1: Yes

Reviewer #2: (No Response)

Reviewer #4: Yes

3. Has the statistical analysis been performed appropriately and rigorously?

Reviewer #1: Yes

Reviewer #2: (No Response)

Reviewer #4: Yes

4. Have the authors made all data underlying the findings in their manuscript fully available?

Reviewer #1: Yes

Reviewer #2: (No Response)

Reviewer #4: Yes

5. Is the manuscript presented in an intelligible fashion and written in standard English?

Reviewer #1: Yes

Reviewer #2: (No Response)

Reviewer #4: Yes

Reviewer #1: There is a significant discrepancy concerning the ethics committee approval number that must be resolved prior to publication. This is a critical issue for research integrity and reproducibility.

Inconsistent Ethics Approval Numbers: In the "Ethics Statement" section on page 4, the approval number is listed as GKSTY-2024032. However, in the main body of the text, under section "2.1 Survey Subject" on page 15, the approval number is cited as GKSTY-2023056.

This inconsistency in fundamental research documentation is a serious concern. The authors must verify the correct approval number and ensure it is stated consistently throughout the final version of the manuscript.

Reviewer #2: The authors have comprehensively and satisfactorily addressed the comments from the reviewers and the editor, which is a key indicator of the scientific rigor required by the journal. The manuscript presents original, methodologically sound research with precise and well-discussed results, and it meets all ethical and data transparency requirements. Therefore, yes, the manuscript is of sufficient quality for publication, as it demonstrates a high technical standard and complies with the journal's scientific and ethical standards.

Reviewer #4: Thank you for an attentive and high‑quality revision. You have addressed the key issues from the previous round: the funding and data‑availability disclosures are now clear, the interpretation reflects the correlational nature of a cross‑sectional design, the path labeling in the Results has been corrected, and the practical implications are more concrete. These changes substantially strengthen methodological transparency and alignment with PLOS ONE’s criteria for publication and data policy.

**Do you want your identity to be public for this peer review?** For information about this choice, including consent withdrawal, please see our Privacy Policy

Reviewer #1: No

Reviewer #2: No

Reviewer #4: No

---

## [Author Response · Author response to Decision Letter 2]

13 Oct 2025

Dear reviewer:

Thank you very much for your meticulous review. Here are my answers to your questions and the relevant revisions to the article. The modified parts are marked in red in the article.

Response to Academic Editor

1.If the reviewer comments include a recommendation to cite specific previously published works, please review and evaluate these publications to determine whether they are relevant and should be cited. There is no requirement to cite these works unless the editor has indicated otherwise. Please review your reference list to ensure that it is complete and correct. If you have cited papers that have been retracted, please include the rationale for doing so in the manuscript text, or remove these references and replace them with relevant current references. Any changes to the reference list should be mentioned in the rebuttal letter that accompanies your revised manuscript. If you need to cite a retracted article, indicate the article’s retracted status in the References list and also include a citation and full reference for the retraction notice.

Response:We have made modifications and marked them in red in the reference.

Response to Additional Editor Comments

1.The sample is restricted to universities in Guangxi Province, which may limit external validity. This should be explicitly stated in both the Abstract and Conclusion to avoid overgeneralization to all undergraduates.

Response: We have added relevant content.

“However, as the sample was drawn exclusively from university students in Guangxi Province, the generalizability of the findings is limited. Future studies should expand the sample scope to include a more diverse population”Line 30-32.“However, since the study merely incorporated university students from Guangxi Province, the generalizability of the findings is constrained”Line 386-387.

2.(Exercise Behavior vs. Exercise Adherence):While measured with different instruments, the conceptual distinction between these constructs could be further clarified in the Methods.

Response: We have added relevant content.

“Exercise behavior refers to an individual's engagement in physical activities. Exercise adherence denotes the ability and willingness to maintain such behavior over an extended period. In essence, it represents the sustained manifestation of exercise behavior” Line 178-181.

3.The manuscript applies both Social Cognitive Theory and the Health Belief Model but largely in parallel. Greater emphasis on how their integration offers novel explanatory power would strengthen the contribution.

Response: We have added relevant content.

“The effective integration of the two provides a more comprehensive explanation of the cognitive-behavioral mechanisms underlying individual actions. The establishment of exercise commitment reflects an individual's stronger pursuit of and belief in health, while health belief, in turn, effectively promotes positive social-cognitive processes that reinforce exercise behavior” Line 84-88.

4.Recommendations such as “incentive mechanisms” remain vague. Expanding this section with concrete examples (e.g., digital self-monitoring tools, peer-led initiatives, or curriculum-based interventions) would enhance applied value.

Response: We have added relevant content.

“For example, implementing diversified sports programs, using digital self-monitoring tools to identify and reward students who maintain regular exercise, establishing a campus competition system with weekly sporting events, incorporating exercise duration into course assessments, and continuously providing positive reinforcement along with moderately challenging physical tasks can all enhance students' enjoyment of exercise”Line 296-301.

5.Some foundational citations are dated (e.g., Glanz, 1997). Consider balancing with more recent literature (2022–2024) to reflect current advances. Read and cite https://doi.org/10.1186/s40359-025-02767-0 ;https://doi.org/10.1108/QEA-06-2024-0054.

Response:We have made modifications and marked them in red in the reference.

Response to Reviewers

1.There is a significant discrepancy concerning the ethics committee approval number that must be resolved prior to publication. This is a critical issue for research integrity and reproducibility.Inconsistent Ethics Approval Numbers: In the "Ethics Statement" section on page 4, the approval number is listed as GKSTY-2024032. However, in the main body of the text, under section "2.1 Survey Subject" on page 15, the approval number is cited as GKSTY-2023056.

Response:We have made modifications.

“(Approval No: GKSTY-2024032)”Line173.

---

## [Editor Report · Decision Letter 2]

5 Nov 2025

Influence of Exercise Commitment on Exercise Adherence Among Undergraduate Students: Chain Mediation Role of Health Beliefs and Exercise Behavior

PONE-D-25-28166R2

Dear Dr. song,

We’re pleased to inform you that your manuscript has been judged scientifically suitable for publication and will be formally accepted for publication once it meets all outstanding technical requirements.

Kind regards,

Nadia Rehman, Ph.D.

Academic Editor

PLOS ONE
---

## [Editor Report · Acceptance letter]

PONE-D-25-28166R2

PLOS ONE

Dear Dr. song,

I'm pleased to inform you that your manuscript has been deemed suitable for publication in PLOS ONE. Congratulations! Your manuscript is now being handed over to our production team.

Kind regards,

on behalf of

Dr. Nadia Rehman

Academic Editor

PLOS ONE